# Re-Pair in Small Space †

**Dominik Köppl** [1,*] **, Tomohiro I** [2] **, Isamu Furuya** [3] **, Yoshimasa Takabatake** [2] **, Kensuke Sakai** [2]
**and Keisuke Goto** [4]

1   M&D Data Science Center, Tokyo Medical and Dental University, Tokyo 113-8510, Japan
2   Kyushu Institute of Technology, Fukuoka 820-8502, Japan; tomohiro@ai.kyutech.ac.jp (T.I.);
    takabatake@ai.kyutech.ac.jp (Y.T.); k_sakai@donald.ai.kyutech.ac.jp (K.S.)
3   Graduate School of IST, Hokkaido University, Hokkaido 060-0814, Japan; furuya@ist.hokudai.ac.jp
4   Fujitsu Laboratories Ltd., Kawasaki 211-8588, Japan; goto.keisuke@fujitsu.com
*   Correspondence: koeppl.dsc@tmd.ac.jp; Tel.: +81-3-5280-8626
†   This paper is an extended version of our paper published in the Prague Stringology Conference 2020: Prague,
    Czech Republic, 31 August–2 September 2020 and at the Data Compression Conference 2020: Virtual
    Conference, 24–27 March 2020.

**Abstract:** Re-Pair is a grammar compression scheme with favorably good compression rates. The computation of Re-Pair comes with the cost of maintaining large frequency tables, which makes it hard to compute Re-Pair on large-scale data sets. As a solution for this problem, we present, given a text of length $n$ whose characters are drawn from an integer alphabet with size $\sigma = n^{\mathcal{O}(1)}$, an $\mathcal{O}(\min(n^2, n^2 \lg \log_\tau n \lg \lg \lg n / \log_\tau n))$ time algorithm computing Re-Pair with $\max((n/c) \lg n, n\lceil \lg \tau \rceil) + \mathcal{O}(\lg n)$ bits of working space including the text space, where $c \geq 1$ is a fixed user-defined constant and $\tau$ is the sum of $\sigma$ and the number of non-terminals. We give variants of our solution working in parallel or in the external memory model. Unfortunately, the algorithm seems not practical since a preliminary version already needs roughly one hour for computing Re-Pair on one megabyte of text.

**Keywords:** grammar compression; Re-Pair; computation in small space; broadword techniques





## 1. Introduction

Re-Pair [1] is a grammar deriving a single string. It is computed by replacing the most frequent bigram in this string with a new non-terminal, recursing until no bigram occurs more than once. Despite this simple-looking description, both the merits and the computational complexity of Re-Pair are intriguing. As a matter of fact, Re-Pair is currently one of the most well-understood grammar schemes.

Besides the seminal work of Larsson and Moffat [1], there are a couple of articles devoted to the compression aspects of Re-Pair: Given a text $T$ of length $n$ whose characters are drawn from an integer alphabet of size $\sigma := n^{\mathcal{O}(1)}$, the output of Re-Pair applied to $T$ is at most $2nH_k(T) + o(n \lg \sigma)$ bits with $k = o(\log_\sigma n)$ when represented naively as a list of character pairs [2], where $H_k$ denotes the empirical entropy of the $k$-th order. Using the encoding of Kieffer and Yang [3], Ochoa and Navarro [4] could improve the output size to at most $nH_k(T) + o(n \lg \sigma)$ bits. Other encodings were recently studied by Ganczorz [5]. Since Re-Pair is a so-called irreducible grammar, its grammar size, i.e., the sum of the symbols on the right-hand side of all rules, is upper bounded by $\mathcal{O}(n/ \log_\sigma n)$ ([3], Lemma 2), which matches the information-theoretic lower bound on the size of a grammar for a string of length $n$. Comparing this size with the size of the smallest grammar, its approximation ratio has $\mathcal{O}((n/ \lg n)^{2/3})$ as an upper bound [6] and $\Omega(\lg n/ \lg \lg n)$ as a lower bound [7]. On the practical side, Yoshida and Kida [8] presented an efficient fixed-length code for compressing the Re-Pair grammar.

Although conceived of as a grammar for compressing texts, Re-Pair has been successfully applied for compressing trees [9], matrices [10], or images [11]. For different

settings or for better compression rates, there is a great interest in modifications to Re-Pair. Charikar et al. [6] (Section G) gave an easy variation to improve the size of the grammar. Another variant, proposed by Claude and Navarro [12], runs in a user-defined working space ($>n \lg n$ bits) and shares with our proposed solution the idea of a table that (a) is stored with the text in the working space and (b) grows in rounds. The variant of González et al. [13] is specialized to compressing a delta-encoded array of integers (i.e., by the differences of subsequent entries). Sekine et al. [14] provided an adaptive variant whose algorithm divides the input into blocks and processes each block based on the rules obtained from the grammars of its preceding blocks. Subsequently, Masaki and Kida [15] gave an online algorithm producing a grammar mimicking Re-Pair. Ganczorz and Jez [16] modified the Re-Pair grammar by disfavoring the replacement of bigrams that cross Lempel–Ziv-77 (LZ77) [17] factorization borders, which allowed the authors to achieve practically smaller grammar sizes. Recently, Furuya et al. [18] presented a variant, called MR-Re-Pair, in which a most frequent maximal repeat is replaced instead of a most frequent bigram.

*1.1. Related Work*

In this article, we focus on the problem of computing the grammar with an algorithm working in text space, forming a bridge between the domain of in-place string algorithms, low-memory compression algorithms, and the domain of Re-Pair computing algorithms. We briefly review some prominent achievements in both domains:

In-place string algorithms: For the LZ77 factorization, Kärkkäinen et al. [19] presented an algorithm computing this factorization with $\mathcal{O}(n/d)$ words on top of the input space in $\mathcal{O}(dn)$ time for a variable $d \geq 1$, achieving $\mathcal{O}(1)$ words with $\mathcal{O}(n^2)$ time. For the suffix sorting problem, Goto [20] gave an algorithm to compute the suffix array [21] with $\mathcal{O}(\lg n)$ bits on top of the output in $\mathcal{O}(n)$ time if each character of the alphabet is present in the text. This condition was improved to alphabet sizes of at most $n$ by Li et al. [22]. Finally, Crochemore et al. [23] showed how to transform a text into its Burrows–Wheeler transform by using $\mathcal{O}(\lg n)$ of additional bits. Due to da Louza et al. [24], this algorithm was extended to compute simultaneously the longest common prefix (LCP) array [21] with $\mathcal{O}(\lg n)$ bits of additional working space.

Low-memory compression algorithms: Simple compression algorithms like run-length compression can be computed in-place and online on the text in linear time. However, a similar result for LZ77 is unknown: A trivial algorithm working with constant number of words (omitting the input text) computes an LZ77 factor starting at $T[i..]$ by linearly scanning $T[1..i-1]$ for the longest previous occurrence $T[j..j+\ell-1] = T[i..i+\ell-1]$ for $j < i$, thus taking quadratic time. A trade-off was proposed by Kärkkäinen et al. [19], who needed $\mathcal{O}(n \lg n/d)$ bits of working space and $\mathcal{O}(nd \lg \lg_n \sigma)$ time for a selectable parameter $d \geq 1$. For the particular case of $d = \epsilon^{-1} \lg n$ for an arbitrary constant $\epsilon > 0$, Kosolobov [25] could improve the running time to $\mathcal{O}(n(\lg \sigma + \lg((\lg n)/\epsilon))/\epsilon)$ for the same space of $\mathcal{O}(\epsilon n)$ bits. Unfortunately, we are unaware of memory-efficient algorithms computing other grammars such as longest-first substitution (LFS) [26], where a modifiable suffix tree is used for computation.

Re-Pair computation: Re-Pair is a grammar proposed by Larsson and Moffat [1], who presented an algorithm computing it in expected linear time with $5n + 4\sigma^2 + 4\sigma' + \sqrt{n}$ words of working space, where $\sigma'$ is the number of non-terminals (produced by Re-Pair). González et al. [13] (Section 4.1) gave another linear time algorithm using $12n + \mathcal{O}(p)$ bytes of working space, where $p$ is the maximum number of distinct bigrams considered at any time. The large space requirements got significantly improved by Bille et al. [27], who presented a randomized linear time algorithm taking $(1 + \epsilon)n + \sqrt{n}$ words on top of the rewritable text space for a constant $\epsilon$ with $0 < \epsilon \leq 1$. Subsequently, they improved their algorithm in [28] to include the text space within the $(1 + \epsilon)n + \sqrt{n}$ words of the working space. However, they assumed that the alphabet size $\sigma$ was constant and $\lceil \lg \sigma \rceil \leq w/2$, where $w$ is the machine word size. They also provided a solution for $\epsilon = 0$ running in expected linear time. Recently, Sakai et al. [29] showed how to convert an arbitrary

grammar (representing a text) into the Re-Pair grammar in compressed space, i.e., without decompressing the text. Combined with a grammar compression that can process the text in compressed space in a streaming fashion, this result leads to the first Re-Pair computation in compressed space.

In a broader picture, Carrascosa et al. [30] provided a generalization called iterative repeat replacement (IRR) , which iteratively selects a substring for replacement via a scoring function. Here, Re-Pair and its variant MR-Re-Pair are specializations of the provided grammar IRR-MO (IRR with maximal number of occurrences)selecting one of the most frequent substrings that have a reoccurring non-overlapping occurrence. (As with bigrams, we only count the number of non-overlapping occurrences.)

### 1.2. Our Contribution

In this article, we propose an algorithm that computes the Re-Pair grammar in $\mathcal{O}(\min(n^2, n^2 \lg \log_\tau n \lg \lg \lg n / \log_\tau n))$ time (cf. Theorems 1 and 2) with $\max((n/c) \lg n, n\lceil \lg \tau \rceil) + \mathcal{O}(\lg n)$ bits of working space including the text space, where $c \geq 1$ is a fixed user-defined constant and $\tau$ is the sum of the alphabet size $\sigma$ and the number of non-terminals $\sigma'$.

We can also compute the byte pair encoding [31], which is Re-Pair with the additional restriction that the algorithm terminates before $\lceil \lg \tau \rceil = \lceil \lg \sigma \rceil$ no longer holds. Hence, we can replace $\tau$ with $\sigma$ in the above space and time bounds.

Given that the characters of the text are drawn from a large integer alphabet with size $\sigma = \Omega(n)$ the algorithm works in-place. (We consider the alphabet as not effective, i.e., a character does not have to appear in the text, as this is a common setting in Unicode texts such as Japanese text. For instance, $n^2 = \Omega(n) \cap n^{\mathcal{O}(1)} \neq \varnothing$ could be such an alphabet size.) In this setting, we obtain the first non-trivial in-place algorithm, as a trivial approach on a text $T$ of length $n$ would compute the most frequent bigram in $\Theta(n^2)$ time by computing the frequency of each bigram $T[i]T[i+1]$ for every integer $i$ with $1 \leq i \leq n-1$, keeping only the most frequent bigram in memory. This sums up to $\mathcal{O}(n^3)$ total time and can be $\Theta(n^3)$ for some texts since there can be $\Theta(n)$ different bigrams considered for replacement by Re-Pair.

To achieve our goal of $\mathcal{O}(n^2)$ total time, we first provide a trade-off algorithm (cf. Lemma 2) finding the $d$ most frequent bigrams in $\mathcal{O}(n^2 \lg d/d)$ time for a trade-off parameter $d$. We subsequently run this algorithm for increasing values of $d$ and show that we need to run it $\mathcal{O}(\lg n)$ times, which gives us $\mathcal{O}(n^2)$ time if $d$ is increasing sufficiently fast. Our major tools are appropriate text partitioning, elementary scans, and sorting steps, which we visualize in Section 2.5 by an example and practically evaluate in Section 2.6. When $\tau = o(n)$, a different approach using word-packing and bit-parallel techniques becomes attractive, leading to an $\mathcal{O}(n^2 \lg \log_\tau n \lg \lg \lg n / \log_\tau n)$ time algorithm, which we explain in Section 3. Our algorithm can be parallelized (Section 5), used in external memory (Section 6), or adapted to compute the MR-Re-Pair grammar in small space (Section 4). Finally, in Section 7, we study several heuristics that make the algorithm faster on specific texts.

### 1.3. Preliminaries

We use the word RAM model with a word size of $\Omega(\lg n)$ for an integer $n \geq 1$. We work in the restore model [32], in which algorithms are allowed to overwrite the input, as long as they can restore the input to its original form.

Strings: Let $T$ be a text of length $n$ whose characters are drawn from an integer alphabet $\Sigma$ of size $\sigma = n^{\mathcal{O}(1)}$. A bigram is an element of $\Sigma^2$. The *frequency* of a bigram $B$ in $T$ is the number of non-overlapping occurrences of $B$ in $T$, which is at most $|T|/2$. For instance, the frequency of the bigram $\mathtt{aa} \in \Sigma^2$ in the text $T = \mathtt{a} \cdots \mathtt{a}$ consisting of $n$ a's is $\lfloor n/2 \rfloor$.

Re-Pair: We reformulate the recursive description in the Introduction by dividing a Re-Pair construction algorithm into turns. Stipulating that $T_i$ is the text after the $i$-th turn

with $i \geq 1$ and $T_0 := T \in \Sigma_0^+$ with $\Sigma_0 := \Sigma$, Re-Pair replaces one of the most frequent bigrams (ties are broken arbitrarily) in $T_{i-1}$ with a non-terminal in the $i$-th turn. Given this bigram is bc $\in \Sigma_{i-1}^2$, Re-Pair replaces all occurrences of bc with a new non-terminal $X_i$ in $T_{i-1}$ and sets $\Sigma_i := \Sigma_{i-1} \cup \{X_i\}$ with $\sigma_i := |\Sigma_i|$ to produce $T_i \in \Sigma_i^+$. Since $|T_i| \leq |T_{i-1}| - 2$, Re-Pair terminates after $m < n/2$ turns such that $T_m \in \Sigma_m^+$ contains no bigram occurring more than once.

## 2. Sequential Algorithm

A major task for producing the Re-Pair grammar is to count the frequencies of the most frequent bigrams. Our work horse for this task is a frequency table. A *frequency table* in $T_i$ of length $f$ stores pairs of the form (bc, $x$), where bc is a bigram and $x$ the frequency of bc in $T_i$. It uses $f \lceil \lg(\sigma_i^2 n_i/2) \rceil$ bits of space since an entry stores a bigram consisting of two characters from $\Sigma_i$ and its respective frequency, which can be at most $n_i/2$. Throughout this paper, we use an elementary in-place sorting algorithm like heapsort:

**Lemma 1** ([33]). *An array of length n can be sorted in-place in $\mathcal{O}(n \lg n)$ time.*

### 2.1. Trade-Off Computation

Using the frequency tables, we present a solution with a trade-off parameter:

**Lemma 2.** *Given an integer d with $d \geq 1$, we can compute the frequencies of the d most frequent bigrams in a text of length n whose characters are drawn from an alphabet of size $\sigma$ in $\mathcal{O}(\max(n, d)n \lg d/d)$ time using $2d \lceil \lg(\sigma^2 n/2) \rceil + \mathcal{O}(\lg n)$ bits.*

**Proof.** Our idea is to partition the set of all bigrams appearing in $T$ into $\lceil n/d \rceil$ subsets, compute the frequencies for each subset, and finally, merge these frequencies. In detail, we partition the text $T = S_1 \cdots S_{\lceil n/d \rceil}$ into $\lceil n/d \rceil$ substrings such that each substring has length $d$ (the last one has a length of at most $d$). Subsequently, we extend $S_j$ to the left (only if $j > 1$) such that $S_j$ and $S_{j+1}$ overlap by one text position, for $1 \leq j < \lceil n/d \rceil$. By doing so, we take the bigram on the border of two adjacent substrings $S_j$ and $S_{j+1}$ for each $j < \lceil n/d \rceil$ into account. Next, we create two frequency tables $F$ and $F'$, each of length $d$ for storing the frequencies of $d$ bigrams. These tables are at the beginning empty. In what follows, we fill $F$ such that after processing $S_i$, $F$ stores the most frequent $d$ bigrams among those bigrams occurring in $S_1, \ldots, S_i$ while $F'$ acts as a temporary space for storing candidate bigrams that can enter $F$.

With $F$ and $F'$, we process each of the $n/d$ substrings $S_j$ as follows: Let us fix an integer $j$ with $1 \leq j \leq \lceil n/d \rceil$. We first put all bigrams of $S_j$ into $F'$ in lexicographic order. We can perform this within the space of $F'$ in $\mathcal{O}(d \lg d)$ time since there are at most $d$ different bigrams in $S_j$. We compute the frequencies of all these bigrams in the complete text $T$ in $\mathcal{O}(n \lg d)$ time by scanning the text from left to right while locating a bigram in $F'$ in $\mathcal{O}(\lg d)$ time with a binary search. Subsequently, we interpret $F$ and $F'$ as one large frequency table, sort it with respect to the frequencies while discarding duplicates such that $F$ stores the $d$ most frequent bigrams in $T[1..jd]$. This sorting step can be done in $\mathcal{O}(d \lg d)$ time. Finally, we clear $F'$ and are done with $S_j$. After the final merge step, we obtain the $d$ most frequent bigrams of $T$ stored in $F$.

Since each of the $\mathcal{O}(n/d)$ merge steps takes $\mathcal{O}(d \lg d + n \lg d)$ time, we need: $\mathcal{O}(\max(d, n) \cdot (n \lg d)/d)$ time. For $d \geq n$, we can build a large frequency table and perform one scan to count the frequencies of all bigrams in $T$. This scan and the final sorting with respect to the counted frequencies can be done in $\mathcal{O}(n \lg n)$ time. □

### 2.2. Algorithmic Ideas

With Lemma 2, we can compute $T_m$ in $\mathcal{O}(mn^2 \lg d/d)$ time with additional $2d \lceil \lg(\sigma_m^2 n/2) \rceil$ bits of working space on top of the text for a parameter $d$ with $1 \leq d \leq n$. (The variable $\tau$

used in the abstract and in the introduction is interchangeable with $\sigma_m$, i.e., $\tau = \sigma_m$.) In the following, we show how this leads us to our first algorithm computing Re-Pair:

**Theorem 1.** *We can compute Re-Pair on a string of length n in $\mathcal{O}(n^2)$ time with $\max((n/c) \lg n, n\lceil \lg \tau \rceil) + \mathcal{O}(\lg n)$ bits of working space including the text space as a rewritable part in the working space, where $c \geq 1$ is a fixed constant and $\tau = \sigma_m$ is the sum of the alphabet size $\sigma$ and the number of non-terminal symbols.*

In our model, we assume that we can enlarge the text $T_i$ from $n_i \lceil \lg \sigma_i \rceil$ bits to $n_i \lceil \lg \sigma_{i+1} \rceil$ bits without additional extra memory. Our main idea is to store a growing frequency table using the space freed up by replacing bigrams with non-terminals. In detail, we maintain a frequency table $F$ in $T_i$ of length $f_k$ for a growing variable $f_k$, which is set to $f_0 := \mathcal{O}(1)$ in the beginning. The table $F$ takes $f_k \lceil \lg(\sigma_i^2 n/2) \rceil$ bits, which is $\mathcal{O}(\lg(\sigma^2 n)) = \mathcal{O}(\lg n)$ bits for $k = 0$. When we want to query it for a most frequent bigram, we linearly scan $F$ in $\mathcal{O}(f_k) = \mathcal{O}(n)$ time, which is not a problem since (a) the number of queries is $m \leq n$ and (b) we aim for $\mathcal{O}(n^2)$ as the overall running time. A consequence is that there is no need to sort the bigrams in $F$ according to their frequencies, which simplifies the following discussion.

Frequency table $F$: With Lemma 2, we can compute $F$ in $\mathcal{O}(n \max(n, f_k) \lg f_k / f_k)$ time. Instead of recomputing $F$ on every turn $i$, we want to recompute it only when it no longer stores a most frequent bigram. However, it is not obvious when this happens as replacing a most frequent bigram during a turn (a) removes this entry in $F$ and (b) can reduce the frequencies of other bigrams in $F$, making them possibly less frequent than other bigrams not tracked by $F$. Hence, the variable $i$ for the $i$-th turn (creating the $i$-th non-terminal) and the variable $k$ for recomputing the frequency table $F$ the $(k + 1)$-st time are loosely connected. We group together all turns with the same $f_k$ and call this group the *k-th round* of the algorithm. At the beginning of each round, we enlarge $f_k$ and create a new $F$ with a capacity for $f_k$ bigrams. Since a recomputation of $F$ takes much time, we want to end a round only if $F$ is no longer useful, i.e., when we no longer can guarantee that $F$ stores a most frequent bigram. To achieve our claimed time bounds, we want to assign all $m$ turns to $\mathcal{O}(\lg n)$ different rounds, which can only be done if $f_k$ grows sufficiently fast.

Algorithm outline: At the beginning of the $k$-th round and the $i$-th turn, we compute the frequency table $F$ storing $f_k$ bigrams and keep additionally the lowest frequency of $F$ as a threshold $t_k$, which is treated as a constant during this round. During the computation of the $i$-th turn, we replace the most frequent bigram (say, bc $\in \Sigma_i^2$) in the text $T_i$ with a non-terminal $X_{i+1}$ to produce $T_{i+1}$. Thereafter, we remove bc from $F$ and update those frequencies in $F$, which were decreased by the replacement of bc with $X_{i+1}$ and add each bigram containing the new character $X_{i+1}$ into $F$ if its frequency is at least $t_k$. Whenever a frequency in $F$ drops below $t_k$, we discard it. If $F$ becomes empty, we move to the $(k + 1)$-st round and create a new $F$ for storing $f_{k+1}$ frequencies. Otherwise ($F$ still stores an entry), we can be sure that $F$ stores a most frequent bigram. In both cases, we recurse with the $(i + 1)$-st turn by selecting the bigram with the highest frequency stored in $F$. We show in Algorithm 1 the pseudo code of this outlined algorithm. We describe in the following how we update $F$ and how large $f_{k+1}$ can become at least.

---

**Algorithm 1:** Algorithmic outline of our proposed algorithm working on a text $T$ with a growing frequency table $F$. The constants $\alpha_i$ and $\beta_i$ are explained in Section 2.3. The same section shows that the outer while loop is executed $\mathcal{O}(\lg n)$ times.

---

1 $k \leftarrow 0, i \leftarrow 0$
2 $f_0 \leftarrow \mathcal{O}(1)$
3 $T_0 \leftarrow T$
4 **while** *highest frequency of a bigram in T is greater than one* **do**　　　　　　▷ during the $k$-th round
5 　｜ $F \leftarrow$ frequency table of Lemma 2 with $d := f_k$
6 　｜ $t_k \leftarrow$ minimum frequency stored in $F$
7 　｜ **while** $F \neq \emptyset$ **do**　　　　　　　　　　　　　　　　　　▷ during the $i$-th turn
8 　｜　｜ bc $\leftarrow$ most frequent bigram stored in $F$
9 　｜　｜ $T_{i+1} \leftarrow T_i.\text{replace}(\text{bc}, X_{i+1})$　　　　　　　　▷ create rule $X_{i+1} \to$ bc
10 　｜　｜ $i \leftarrow i + 1$　　　　　　　　　　　　　　▷ introduce the $(i+1)$-th turn
11 　｜　｜ remove all bigrams with frequency lower than $t_k$ from $F$
12 　｜　｜ add new bigrams to $F$ having $X_i$ as left or right character and a frequency of at least $t_k$
13 　｜ $f_{k+1} \leftarrow f_k + \max(2/\beta_i, (f_k - 1)/(2\beta_i))/\alpha_i$
14 　｜ $k \leftarrow k + 1$　　　　　　　　　　　　　　　▷ introduce the $(k+1)$-th round
15 Invariant: $i = m$ (the number of non-terminals)

---

### 2.3. Algorithmic Details

Suppose that we are in the $k$-th round and in the $i$-th turn. Let $t_k$ be the lowest frequency in $F$ computed at the beginning of the $k$-th round. We keep $t_k$ as a constant threshold for the invariant that all frequencies in $F$ are at least $t_k$ during the $k$-th round. With this threshold, we can assure in the following that $F$ is either empty or stores a most frequent bigram.

Now suppose that the most frequent bigram of $T_i$ is bc $\in \Sigma_i^2$, which is stored in $F$. To produce $T_{i+1}$ (and hence advancing to the $(i+1)$-st turn), we enlarge the space of $T_i$ from $n_i \lceil \lg \sigma_i \rceil$ to $n_i \lceil \lg \sigma_{i+1} \rceil$ and replace all occurrences of bc in $T_i$ with a new non-terminal $X_{i+1}$. Subsequently, we would like to take the next bigram of $F$. For that, however, we need to update the stored frequencies in $F$. To see this necessity, suppose that there is an occurrence of abcd with two characters a, d $\in \Sigma_i$ in $T_i$. By replacing bc with $X_{i+1}$,

1. the frequencies of ab and cd decrease by one (for the border case a = b = c (resp. b = c = d), there is no need to decrement the frequency of ab (resp. cd)), and
2. the frequencies of a$X_{i+1}$ and $X_{i+1}$d increase by one.

Updating $F$. We can take care of the former changes (1) by decreasing the respective bigram in $F$ (in the case that it is present). If the frequency of this bigram drops below the threshold $t_k$, we remove it from $F$ as there may be bigrams with a higher frequency that are not present in $F$. To cope with the latter changes (2), we track the characters adjacent to $X_{i+1}$ after the replacement, count their numbers, and add their respective bigrams to $F$ if their frequencies are sufficiently high. In detail, suppose that we have substituted bc with $X_{i+1}$ exactly $h$ times. Consequently, with the new text $T_{i+1}$ we have additionally $h \lg \sigma_{i+1}$ bits of free space (the free space is consecutive after shifting all characters to the left), which we call $D$ in the following. Subsequently, we scan the text and put the characters of $\Sigma_{i+1}$ appearing to the left of each of the $h$ occurrences of $X_{i+1}$ into $D$. After sorting the characters in $D$ lexicographically, we can count the frequency of a$X_{i+1}$ for each character a $\in \Sigma_{i+1}$ preceding an occurrence of $X_{i+1}$ in the text $T_{i+1}$ by scanning $D$ linearly. If the obtained frequency of such a bigram a$X_{i+1}$ is at least as high as the threshold $t_k$, we insert a$X_{i+1}$ into $F$ and subsequently discard a bigram with the currently lowest frequency in $F$ if the size of $F$ has become $f_k + 1$. In the case that we visit a run of $X_{i+1}$'s during the creation of $D$, we must take care of not counting the overlapping occurrences of $X_{i+1}X_{i+1}$. Finally, we can count analogously the occurrences of $X_{i+1}$d for all characters d $\in \Sigma_i$ succeeding an occurrence of $X_{i+1}$.

Capacity of $F$: After the above procedure, we update the frequencies of $F$. When $F$ becomes empty, all bigrams stored in $F$ are replaced or have a frequency that becomes less than $t_k$. Subsequently, we end the $k$-th round and continue with the $(k+1)$-st round by (a) creating a new frequency table $F$ with capacity $f_{k+1}$ and (b) setting the new threshold $t_{k+1}$ to the minimal frequency in $F$. In what follows, we (a) analyze in detail when $F$ becomes empty (as this determines the sizes $f_k$ and $f_{k+1}$) and (b) show that we can compensate the number of discarded bigrams with an enlargement of $F$'s capacity from $f_k$ bigrams to $f_{k+1}$ bigrams for the sake of our aimed total running time.

Next, we analyze how many characters we have to free up (i.e., how many bigram occurrences we have to replace) to gain enough space for storing an additional frequency. Let $\delta_i := \lg(\sigma_{i+1}^2 n_i / 2)$ be the number of bits needed to store one entry in $F$, and let $\beta_i := \min(\delta_i / \lg \sigma_{i+1}, c\delta_i / \lg n)$ be the minimum number of characters that need to be freed to store one frequency in this space. To understand the value of $\beta_i$, we look at the arguments of the minimum function in the definition of $\beta_i$ and simultaneously at the maximum function in our aimed working space of $\max(n\lceil \lg \sigma_m \rceil, (n/c) \lg n) + \mathcal{O}(\lg n)$ bits (cf. Theorem 1):

1.　The first item in this maximum function allows us to spend $\lg \sigma_{i+1}$ bits for each freed character such that we obtain space for one additional entry in $F$ after freeing $\delta_i / \lg \sigma_{i+1}$ characters.
2.　The second item allows us to use $\lg n$ additional bits after freeing up $c$ characters. This additional treatment helps us to let $f_k$ grow sufficiently fast in the first steps to save our $\mathcal{O}(n^2)$ time bound, as for sufficiently small alphabets and large text sizes, $\lg(\sigma^2 n/2)/\lg \sigma = \mathcal{O}(\lg n)$, which means that we might run the first $\mathcal{O}(\lg n)$ turns with $f_k = \mathcal{O}(1)$ and, therefore, already spend $\mathcal{O}(n^2 \lg n)$ time. Hence, after freeing up $c\delta_i / \lg n$ characters, we have space to store one additional entry in $F$.

With $\beta_i = \min(\delta_i / \lg \sigma_{i+1}, c\delta_i / \lg n) = \mathcal{O}(\log_\sigma n) \cap \mathcal{O}(\log_n \sigma) = \mathcal{O}(1)$, we have the sufficient condition that replacing a constant number of characters gives us enough space for storing an additional frequency.

If we assume that replacing the occurrences of a bigram stored in $F$ does not decrease the other frequencies stored in $F$, the analysis is now simple: Since each bigram in $F$ has a frequency of at least two, $f_{k+1} \geq f_k + f_k / \beta_i$. Since $\beta_i = \mathcal{O}(1)$, this lets $f_k$ grow exponentially, meaning that we need $\mathcal{O}(\lg n)$ rounds. In what follows, we show that this is also true in the general case.

**Lemma 3.** *Given that the frequency of all bigrams in F drops below the threshold $t_k$ after replacing the most frequent bigram* bc, *then its frequency has to be at least* $\max(2, |F| - 1/2)$, *where* $|F| \leq f_k$ *is the number of frequencies stored in F.*

**Proof.** If the frequency of bc in $T_i$ is $x$, then we can reduce at most $2x$ frequencies of other bigrams (both the left character and the right character of each occurrence of bc can contribute to an occurrence of another bigram). Since a bigram must occur at least twice in $T_i$ to be present in $F$, the frequency of bc has to be at least $\max(2, (f_k - 1)/2)$ for discarding all bigrams of $F$. □

Suppose that we have enough space available for storing the frequencies of $\alpha_i f_k$ bigrams, where $\alpha_i$ is a constant (depending on $\sigma_i$ and $n_i$) such that $F$ and the working space of Lemma 2 with $d = f_k$ can be stored within this space. With $\beta_i$ and Lemma 3 with $|F| = f_k$, we have:

$$\alpha_i f_{k+1} = \alpha_i f_k + \max(2/\beta_i, (f_k - 1)/(2\beta_i))$$
$$= \alpha_i f_k \max(1 + 2/(\alpha_i \beta_i f_k), 1 + 1/(2\alpha_i \beta_i) - 1/(2\alpha_i \beta_i f_k))$$
$$\geq \alpha_i f_k (1 + 2/(5\alpha_i \beta_i)) =: \gamma_i \alpha_i f_k \text{ with } \gamma_i := 1 + 2/(5\alpha_i \beta_i),$$

where we use the equivalence $1 + 2/(\alpha_i \beta_i f_k) = 1 + 1/(2\alpha_i \beta_i) - 1/(2\alpha_i \beta_i f_k) \Leftrightarrow 5 = f_k$ to estimate the two arguments of the maximum function.

Since we let $f_k$ grow by a factor of at least $\gamma := \min_{1 \le i \le m} \gamma_i > 1$ for each recomputation of $F$, $f_k = \Omega(\gamma^k)$, and therefore, $f_k = \Theta(n)$ after $k = \mathcal{O}(\lg n)$ steps. Consequently, after reaching $k = \mathcal{O}(\lg n)$, we can iterate the above procedure a constant number of times to compute the non-terminals of the remaining bigrams occurring at least twice.

Time analysis: In total, we have $\mathcal{O}(\lg n)$ rounds. At the start of the $k$-th round, we compute $F$ with the algorithm of Lemma 2 with $d = f_k$ on a text of length at most $n - f_k$ in $\mathcal{O}(n(n - f_k) \cdot \lg f_k / f_k)$ time with $f_k \le n$. Summing this up, we get:

$$\mathcal{O}\left( \sum_{k=0}^{\mathcal{O}(\lg n)} \frac{n - f_k}{f_k} n \lg f_k \right) = \mathcal{O}\left( n^2 \sum_{k}^{\lg n} \frac{k}{\gamma^k} \right) = \mathcal{O}\left( n^2 \right) \text{ time.} \tag{1}$$

In the $i$-th turn, we update $F$ by decreasing the frequencies of the bigrams affected by the substitution of the most frequent bigram bc with $X_{i+1}$. For decreasing such a frequency, we look up its respective bigram with a linear scan in $F$, which takes $f_k = \mathcal{O}(n)$ time. However, since this decrease is accompanied with a replacement of an occurrence of bc, we obtain $\mathcal{O}(n^2)$ total time by charging each text position with $\mathcal{O}(n)$ time for a linear search in $F$. With the same argument, we can bound the total time for sorting the characters in $D$ to $\mathcal{O}(n^2)$ overall time: Since we spend $\mathcal{O}(h \lg h)$ time on sorting $h$ characters preceding or succeeding a replaced character and $\mathcal{O}(f_k) = \mathcal{O}(n)$ time on swapping a sufficiently large new bigram composed of $X_{i+1}$ and a character of $\Sigma_{i+1}$ with a bigram with the lowest frequency in $F$, we charge each text position again with $\mathcal{O}(n)$ time. Putting all time bounds together gives the claim of Theorem 1.

### 2.4. Storing the Output In-Place

Finally, we show that we can store the computed grammar in text space. More precisely, we want to store the grammar in an auxiliary array $A$ packed at the end of the working space such that the entry $A[i]$ stores the right-hand side of the non-terminal $X_i$, which is a bigram. Thus, the non-terminals are represented implicitly as indices of the array $A$. We therefore need to subtract $2 \lg \sigma_i$ bits of space from our working space $\alpha_i f_k$ after the $i$-th turn. By adjusting $\alpha_i$ in the above equations, we can deal with this additional space requirement as long as the frequencies of the replaced bigrams are at least three (we charge two occurrences for growing the space of $A$).

When only bigrams with frequencies of at most two remain, we switch to a simpler algorithm, discarding the idea of maintaining the frequency table $F$: Suppose that we work with the text $T_i$. Let $\lambda$ be a text position, which is one in the beginning, but will be incremented in the following turns while holding the invariant $T[1..\lambda]$ that does not contain a bigram of frequency two. We scan $T_i[\lambda..n]$ linearly from left to right and check, for each text position $j$, whether the bigram $T_i[j]T_i[j + 1]$ has another occurrence $T_i[j']T_i[j' + 1] = T_i[j]T_i[j + 1]$ with $j' > j + 1$, and if so,

(a)     append $T_i[j]T_i[j + 1]$ to $A$,
(b)     replace $T_i[j]T_i[j + 1]$ and $T_i[j']T_i[j' + 1]$ with a new non-terminal $X_{i+1}$ to transform $T_i$ to $T_{i+1}$, and
(c)     recurse on $T_{i+1}$ with $\lambda := j$ until no bigram with frequency two is left.

The position $\lambda$, which we never decrement, helps us to skip over all text positions starting with bigrams with a frequency of one. Thus, the algorithm spends $\mathcal{O}(n)$ time for each such text position and $\mathcal{O}(n)$ time for each bigram with frequency two. Since there are at most $n$ such bigrams, the overall running time of this algorithm is $\mathcal{O}(n^2)$.

**Remark 1** (Pointer machine model). *Refraining from the usage of complicated algorithms, our algorithm consists only of elementary sorting and scanning steps. This allows us to run our algorithm on a pointer machine, obtaining the same time bound of $\mathcal{O}(n^2)$. For the space bounds, we*

*assume that the text is given in n words, where a word is large enough to store an element of $\Sigma_m$ or a text position.*

### 2.5. Step-by-Step Execution

Here, we present an exemplary execution of the first turn (of the first round) on the input $T = $ cabaacabcabaacaaabcab. We visualize each step of this turn as a row in Figure 1. A detailed description of each row follows:

**Row 1:** Suppose that we have computed $F$, which has the constant number of entries $f_0 = 3$ (in the later turns when the size $f_k$ becomes larger, $F$ will be put in the text space). The highest frequency is five achieved by ab and ca. The lowest frequency represented in $F$ is three, which becomes the threshold $t_0$ for a bigram to be present in $F$ such that bigrams whose frequencies drop below $t_0$ are removed from $F$. This threshold is a constant for all later turns until $F$ is rebuilt (in the following round). During Turn 1, the algorithm proceeds now as follows:

**Row 2:** Choose ab as a bigram to replace with a new non-terminal $X_1$ (break ties arbitrarily). Replace every occurrence of ab with $X_1$ while decrementing frequencies in $F$ according to the neighboring characters of the replaced occurrence.

**Row 3:** Remove from $F$ every bigram whose frequency falls below the threshold. Obtain space for $D$ by aligning the compressed text $T_1$ (the process of Row 2 and Row 3 can be done simultaneously).

**Row 4:** Scan the text and copy each character preceding an occurrence of $X_1$ in $T_1$ to $D$.

**Row 5:** Sort characters in $D$ lexicographically.

**Row 6:** Insert new bigrams (consisting of a character of $D$ and $X_1$) whose frequencies are at least as large as the threshold.

**Row 7:** Scan the text again and copy each character succeeding an occurrence of $X_1$ in $T_1$ to $D$ (symmetric to Row 4).

**Row 8:** Sort all characters in $D$ lexicographically (symmetric to Row 5).

**Row 9:** Insert new bigrams whose frequencies are at least as large as the threshold (symmetric to Row 6).

| | 1 | 2 | 3 | 4 | 5 | 6 | 7 | 8 | 9 | 10 | 11 | 12 | 13 | 14 | 15 | 16 | 17 | 18 | 19 | 20 | 21 | 22 | 23 | 24 |
|---|---|---|---|---|---|---|---|---|---|---|---|---|---|---|---|---|---|---|---|---|---|---|---|---|
| 1 | c | a | b | a | a | c | a | b | c | a | b | a | a | c | a | a | a | b | c | a | b | ab:5 | ca:5 | aa:3 |
| 2 | c | $X_1$ | | a | a | c | $X_1$ | | c | $X_1$ | | a | a | c | a | a | $X_1$ | | c | $X_1$ | | ab:0 | ca:1 | aa:3 |
| 3 | c | $X_1$ | a | a | c | $X_1$ | c | $X_1$ | a | a | c | a | a | $X_1$ | c | $X_1$ | | | | | | | | aa:3 |
| 4 | c | $X_1$ | a | a | c | $X_1$ | c | $X_1$ | a | a | c | a | a | $X_1$ | c | $X_1$ | c | c | c | a | c | | | aa:3 |
| 5 | c | $X_1$ | a | a | c | $X_1$ | c | $X_1$ | a | a | c | a | a | $X_1$ | c | $X_1$ | a | c | c | c | c | | | aa:3 |
| 6 | c | $X_1$ | a | a | c | $X_1$ | c | $X_1$ | a | a | c | a | a | $X_1$ | c | $X_1$ | | | | | | c$X_1$:4 | | aa:3 |
| 7 | c | $X_1$ | a | a | c | $X_1$ | c | $X_1$ | a | a | c | a | a | $X_1$ | c | $X_1$ | a | c | a | c | | c$X_1$:4 | | aa:3 |
| 8 | c | $X_1$ | a | a | c | $X_1$ | c | $X_1$ | a | a | c | a | a | $X_1$ | c | $X_1$ | a | a | c | c | | c$X_1$:4 | | aa:3 |
| 9 | c | $X_1$ | a | a | c | $X_1$ | c | $X_1$ | a | a | c | a | a | $X_1$ | c | $X_1$ | | | | | | c$X_1$:4 | | aa:3 |

**Figure 1.** Step-by-step execution of the first turn of our algorithm on the string $T = $ cabaacabcabaacaaabcab. The turn starts with the memory configuration given in Row 1. Positions 1 to 21 are text positions, and Positions 22 to 24 belong to $F$ ($f_0 = 3$, and it is assumed that a frequency fits into a text entry). Subsequent rows depict the memory configuration during Turn 1. A comment on each row is given in Section 2.5.

### 2.6. Implementation

At https://github.com/koeppl/repair-inplace, we provide a simplified implementation in C++17. The simplification is that we (a) fix the bit width of the text space to 16 bit

and (b) assume that $\Sigma$ is the byte alphabet. We further skip the step increasing the bit width of the text from $\lg \sigma_i$ to $\lg \sigma_{i+1}$. This means that the program works as long as the characters of $\Sigma_m$ fit into 16 bits. The benchmark, whose results are displayed in Table 1, was conducted on a Mac Pro Server with an Intel Xeon CPU X5670 clocked at 2.93 GHz running Arch Linux. The implementation was compiled with `gcc-8.2.1` in the highest optimization mode `-O3`. Looking at Table 1, we observe that the running time is super-linear to the input size on all text instances, which we obtained from the Pizza&Chili corpus (http://pizzachili.dcc.uchile.cl/). We conducted the same experiments with an implementation of Gonzalo Navarro (https://users.dcc.uchile.cl/~gnavarro/software/repair.tgz) in Table 2 with considerably better running times while restricting the algorithm to use 1 MB of RAM during execution. Table 3 gives some characteristics about the used data sets. We see that the number of rounds is the number of turns plus one for every unary string $a^{2^k}$ with an integer $k \geq 1$ since the text contains only one bigram with a frequency larger than two in each round. Replacing this bigram in the text makes $F$ empty such that the algorithm recomputes $F$ after each turn. Note that the number of rounds can drop while scaling the prefix length based on the choice of the bigrams stored in $F$.

**Table 1.** Experimental evaluation of our implementation and the implementation of Navarro described in Section 2.6. Table entries are running times in seconds. The last line is the benchmark on the unary string $\mathtt{aa \cdots a}$.

| Data Set | Our Implementation | | | | | Implementation of Navarro | | | | |
|---|---|---|---|---|---|---|---|---|---|---|
| | Prefix Size in KiB | | | | | | | | | |
| | 64 | 128 | 256 | 512 | 1024 | 64 | 128 | 256 | 512 | 1024 |
| ESCHERICHIA_COLI | 20.68 | 130.47 | 516.67 | 1708.02 | 10,112.47 | 0.01 | 0.02 | 0.07 | 0.18 | 0.29 |
| CERE | 13.69 | 90.83 | 443.17 | 2125.17 | 9185.58 | 0.01 | 0.02 | 0.04 | 0.16 | 0.22 |
| COREUTILS | 12.88 | 75.64 | 325.51 | 1502.89 | 5144.18 | 0.01 | 0.05 | 0.05 | 0.14 | 0.29 |
| EINSTEIN.DE.TXT | 19.55 | 88.34 | 181.84 | 805.81 | 4559.79 | 0.01 | 0.04 | 0.08 | 0.10 | 0.25 |
| EINSTEIN.EN.TXT | 21.11 | 78.57 | 160.41 | 900.79 | 4353.81 | 0.01 | 0.02 | 0.05 | 0.21 | 0.51 |
| INFLUENZA | 41.01 | 160.68 | 667.58 | 2630.65 | 10,526.23 | 0.03 | 0.02 | 0.05 | 0.11 | 0.36 |
| KERNEL | 20.53 | 101.84 | 208.08 | 1575.48 | 5067.80 | 0.01 | 0.04 | 0.09 | 0.18 | 0.27 |
| PARA | 20.90 | 175.93 | 370.72 | 2826.76 | 9462.74 | 0.01 | 0.01 | 0.08 | 0.12 | 0.35 |
| WORLD_LEADERS | 11.92 | 21.82 | 167.52 | 661.52 | 1718.36 | 0.01 | 0.01 | 0.06 | 0.11 | 0.25 |
| $\mathtt{aa \cdots a}$ | 0.35 | 0.92 | 3.90 | 14.16 | 61.74 | 0.01 | 0.01 | 0.05 | 0.05 | 0.12 |

**Table 2.** Experimental evaluation of the implementation of Navarro. Table entries are running times in seconds.

| Data Set | Prefix Size in KiB | | | | |
|---|---|---|---|---|---|
| | 64 | 128 | 256 | 512 | 1024 |
| ESCHERICHIA_COLI | 0.01 | 0.02 | 0.07 | 0.18 | 0.29 |
| CERE | 0.01 | 0.02 | 0.04 | 0.16 | 0.22 |
| COREUTILS | 0.01 | 0.05 | 0.05 | 0.14 | 0.29 |
| EINSTEIN.DE.TXT | 0.01 | 0.04 | 0.08 | 0.10 | 0.25 |
| EINSTEIN.EN.TXT | 0.01 | 0.02 | 0.05 | 0.21 | 0.51 |
| INFLUENZA | 0.03 | 0.02 | 0.05 | 0.11 | 0.36 |
| KERNEL | 0.01 | 0.04 | 0.09 | 0.18 | 0.27 |
| PARA | 0.01 | 0.01 | 0.08 | 0.12 | 0.35 |
| WORLD_LEADERS | 0.01 | 0.01 | 0.06 | 0.11 | 0.25 |

**Table 3.** Characteristics of our data sets used in Section 2.6. The number of turns and rounds are given for each of the prefix sizes 128, 256, 512, and 1024 KiB of the respective data sets. The number of turns reflecting the number of non-terminals is given in units of thousands. The turns of the unary string aa···a are in plain units (not divided by thousand).

| Data Set | $\sigma$ | Turns/1000 Prefix Size in KiB | | | | | Rounds Prefix Size in KiB | | | | |
|---|---|---|---|---|---|---|---|---|---|---|---|
| | | $2^6$ | $2^7$ | $2^8$ | $2^9$ | $2^{10}$ | $2^6$ | $2^7$ | $2^8$ | $2^9$ | $2^{10}$ |
| ESCHERICHIA_COLI | 4 | 1.8 | 3.2 | 5.6 | 10.3 | 18.1 | 6 | 9 | 9 | 12 | 12 |
| CERE | 5 | 1.4 | 2.8 | 5.0 | 9.2 | 15.1 | 13 | 14 | 14 | 14 | 14 |
| COREUTILS | 113 | 4.7 | 6.7 | 10.2 | 16.1 | 26.5 | 15 | 15 | 15 | 14 | 14 |
| EINSTEIN.DE.TXT | 95 | 1.7 | 2.8 | 3.7 | 5.2 | 9.7 | 14 | 14 | 15 | 16 | 16 |
| EINSTEIN.EN.TXT | 87 | 3.3 | 3.5 | 3.8 | 4.5 | 8.6 | 16 | 15 | 15 | 15 | 17 |
| INFLUENZA | 7 | 2.5 | 3.7 | 9.5 | 13.4 | 22.1 | 11 | 12 | 14 | 13 | 15 |
| KERNEL | 160 | 4.5 | 8.0 | 13.9 | 24.5 | 43.7 | 10 | 11 | 14 | 14 | 13 |
| PARA | 5 | 1.8 | 3.2 | 5.8 | 10.1 | 17.6 | 12 | 12 | 13 | 13 | 14 |
| WORLD_LEADERS | 87 | 2.6 | 4.3 | 6.1 | 10.0 | 42.1 | 11 | 11 | 11 | 11 | 14 |
| aa···a | 1 | 15 | 16 | 17 | 18 | 19 | 16 | 17 | 18 | 19 | 20 |

## 3. Bit-Parallel Algorithm

In the case that $\tau = \sigma_m$ is $o(n)$ (and therefore, $\sigma = o(n)$), a word-packing approach becomes interesting. We present techniques speeding up the previously introduced operations on chunks of $\mathcal{O}(\log_\tau n)$ characters from $\mathcal{O}(\log_\tau n)$ time to $\mathcal{O}(\lg \lg \lg n)$ time. In the end, these techniques allow us to speed up the sequential algorithm of Theorem 1 from $\mathcal{O}(n^2)$ time to the following:

**Theorem 2.** *We can compute Re-Pair on a string of length n in $\mathcal{O}(n^2 \lg \log_\tau n \lg \lg \lg n / \log_\tau n)$ time with $\max((n/c) \lg n, n\lceil \lg \tau \rceil) + \mathcal{O}(\lg n)$ bits of working space including the text space, where $c \geq 1$ is a fixed constant and $\tau = \sigma_m$ is the sum of the alphabet size $\sigma$ and the number of non-terminal symbols.*

Note that the $\mathcal{O}(\lg \lg \lg n)$ time factor is due to the popcount function [34] (Algorithm 1), which has been optimized to a single instruction on modern computer architectures. Our toolbox consists of several elementary instructions shown in Figure 2. There, $\text{msb}(X)$ can be computed in constant time algorithm using $\mathcal{O}(\lg n)$ bits [35] (Section 5). The last two functions in Figure 2 are explained in Figure 3.

### 3.1. Broadword Search

First, we deal with accelerating the computation of the frequency of a bigram in $T$ by exploiting broadword search thanks to the word RAM model. We start with the search of single characters and subsequently extend this result to bigrams:

| Operation | Description |
|---|---|
| $X \ll j$ | shift $X$ $j$ positions to the left |
| $X \gg j$ | shift $X$ $j$ positions to the right |
| $\neg X$ | bitwise NOT of $X$ |
| $X \otimes Y$ | bitwise XOR of $X$ and $Y$ |
| $-1$ | bit vector consisting only of one bits |
| $\text{msb}(X)$ | returns the position of the most significant set bit of $X$, i.e., $\lfloor \lg X \rfloor + 1$; |
| $\text{rmPreRun}(X)$ | sets all bits of the maximal prefix of consecutive ones to zero |
| $\text{rmSufRun}(X)$ | sets all bits of the maximal suffix of consecutive ones to zero |

**Figure 2.** Operations used in Figures 4 and 5 for two bit vectors $X$ and $Y$. All operations can be computed in constant time. See Figure 3 for an example of rmSufRun and rmPreRun.

| rmPreRun($X$) | | rmSufRun($X$) | |
|---|---|---|---|
| Operation | Example | Operation | Example |
| $X$ | 11100110 | $X$ | 01100111 |
| $\neg X$ | 00011001 | $\neg X$ | 10011000 |
| $1 \ll (1 + \mathrm{msb}(\neg X))$ | 00100000 | $\neg X - 1$ | 10010111 |
| $(1 \ll (1 + \mathrm{msb}(\neg X))) - 1$ | 00011111 | $(\neg X - 1) \,\&\, X$ | 00000111 |
| $((1 \ll (1 + \mathrm{msb}(\neg X))) - 1) \,\&\, X$ | 00000110 | $\neg((\neg X - 1) \,\&\, X)$ | 11111000 |
| | | $\neg((\neg X - 1) \,\&\, X) \,\&\, X$ | 01100000 |

**Figure 3.** Step-by-step execution of rmPreRun($X$) and rmSufRun($X$) introduced in Figure 2 on a bit vector $X$.

**Lemma 4.** *We can count the occurrences of a character $c \in \Sigma$ in a string of length $\mathcal{O}(\log_\sigma n)$ in $\mathcal{O}(\lg \lg \lg n)$ time.*

**Proof.** Let $q$ be the largest multiple of $\lceil \lg \sigma \rceil$ fitting into a computer word, divided by $\lceil \lg \sigma \rceil$. Let $S \in \Sigma^*$ be a string of length $q$. Our first task is to compute a bit mask of length $q \lceil \lg \sigma \rceil$ marking the occurrences of a character $c \in \Sigma$ in $S$ with a '1'. For that, we follow the constant time broadword pattern matching of Knuth [36] (Section 7.1.3); see https://github.com/koeppl/broadwordsearch for a practical implementation. Let $H$ and $L$ be two bit vectors of length $\lceil \lg \sigma \rceil$ having marked only the most significant or the least significant bit, respectively. Let $H^q$ and $L^q$ denote the $q$ times concatenation of $H$ and $L$, respectively. Then, the operations in Figure 4 yield an array $X$ of length $q$ with:

$$X[i] = \begin{cases} 2^{\lceil \lg \sigma \rceil} - 1 & \text{if } S[i] = c, \\ 0 & \text{otherwise,} \end{cases} \tag{2}$$

where each entry of $X$ has $\lceil \lg \sigma \rceil$ bits.

| Operation | Description | | Example |
|---|---|---|---|
| read $S$ | | | 101010000 $\to S$ |
| $X \leftarrow S \otimes c^q$ | match $S$ with $c^q$; $X[i] = 0 \Leftrightarrow$ $S[i] = c$ | $\otimes$ $\equiv$ | 101010000 $= S$ 010010010 111000010 $\to X$ |
| $Y \leftarrow X - L^q$ | | $-$ $\equiv$ | 111000010 $= X$ 001001001 101111001 $\to Y$ |
| $X \leftarrow Y \,\&\, \neg X$ | $X[i] \,\&\, 2^{\lceil \lg \sigma \rceil} - 1 = 1 \Leftrightarrow$ $S[i] = c$ | $\&$ $\equiv$ | 101111001 $= Y$ 000111101 000111001 $\to X$ |
| $X \leftarrow X \,\&\, H^q$ | $X[i] = 0 \Leftrightarrow S[i] \neq c$ | $\&$ $\equiv$ | 000111001 $= X$ 100100100 000100000 $\to X$ |
| $X \leftarrow (X - (X \gg (\lceil \lg \sigma \rceil - 1))) \mid X$ | $X$ as in Equation (2) | $-$ $=$ $\mid$ $=$ | 000100000 $= X$ 000001000 000011000 000100000 000111000 $\to X$ |

**Figure 4.** Matching all occurrences of a character in a string $S$ fitting into a computer word in constant time by using bit-parallel instructions. For the last step, special care has to be taken when the last character of $S$ is a match, as shifting $X$ $\lceil \lg \sigma \rceil$ bits to the right might erase a '1' bit witnessing the rightmost match. In the description column, $X$ is treated as an array of integers with bit width $\lceil \lg \sigma \rceil$. In this example, $S = 101010000$, $c$ has the bit representation 010 with $\lg \sigma = 3$, and $q = 3$.

To obtain the number of occurrences of $c$ in $S$, we use the popcount operation returning the number of zero bits in $X$ and divide the result by $\lceil \lg \sigma \rceil$. The popcount instruction takes $\mathcal{O}(\lg \lg \lg n)$ time ([34] Algorithm 1). □

Having Lemma 4, we show that we can compute the frequency of a bigram in $T$ in $\mathcal{O}(n \lg \lg \lg n / \log_\sigma n)$ time. For that, we interpret $T \in \Sigma^n$ of length $n$ as a text $T \in (\Sigma^2)^{\lceil n/2 \rceil}$ of length $\lceil n/2 \rceil$. Then, we partition $T$ into strings fitting into a computer word and call each string of this partition a *chunk*. For each chunk, we can apply Lemma 4 by treating a bigram $c \in \Sigma^2$ as a single character. The result is, however, not the frequency of the bigram $c$ in general. For computing the frequency a bigram $\mathtt{bc} \in \Sigma^2$, we distinguish the cases $\mathtt{b} \neq \mathtt{c}$ and $\mathtt{b} = \mathtt{c}$.

Case $\mathtt{b} \neq \mathtt{c}$: By applying Lemma 4 to find the character $\mathtt{bc} \in \Sigma^2$ in a chunk $S$ (interpreted as a string of length $\lfloor q/2 \rfloor$ on the alphabet $\Sigma^2$), we obtain the number of occurrences of $\mathtt{bc}$ starting at odd positions in $S$. To obtain this number for all even positions, we apply the procedure to $\mathtt{d}S$ with $\mathtt{d} \in \Sigma \setminus \{\mathtt{b}, \mathtt{c}\}$. Additional care has to be taken at the borders of each chunk matching the last character of the current chunk and the first character of the subsequent chunk with $\mathtt{b}$ and $\mathtt{c}$, respectively.

Case $\mathtt{b} = \mathtt{c}$: This case is more involving as overlapping occurrences of $\mathtt{bb}$ can occur in $S$, which we must not count. To this end, we watch out for *runs* of $\mathtt{b}$'s, i.e., substrings of maximal lengths consisting of the character $\mathtt{b}$ (here, we consider also maximal substrings of $\mathtt{b}$ with length one as a run). We separate these runs into runs ending either at even or at odd positions. We do this because the frequency of $\mathtt{bb}$ in a run of $\mathtt{b}$'s ending at an even (resp. odd) position is the number of occurrences of $\mathtt{bb}$ within this run ending at an even (resp. odd) position. We can compute these positions similarly to the approach for $\mathtt{b} \neq \mathtt{c}$ by first (a) hiding runs ending at even (resp. odd) positions and then (b) counting all bigrams ending at even (resp. odd) positions. Runs of $\mathtt{b}$'s that are a prefix or a suffix of $S$ are handled individually if $S$ is neither the first nor the last chunk of $T$, respectively. That is because a run passing a chunk border starts and ends in different chunks. To take care of those runs, we remember the number of $\mathtt{b}$'s of the longest suffix of every chunk and accumulate this number until we find the end of this run, which is a prefix of a subsequent chunk. The procedure for counting the frequency of $\mathtt{bb}$ inside $S$ is explained with an example in Figure 5. With the aforementioned analysis of the runs crossing chunk borders, we can extend this procedure to count the frequency of $\mathtt{bb}$ in $T$. We conclude:

**Lemma 5.** *We can compute the frequency of a bigram in a string T of length n whose characters are drawn from an alphabet of size $\sigma$ in $\mathcal{O}(n \lg \lg \lg n / \log_\sigma n)$ time.*

*3.2. Bit-Parallel Adaption*

Similarly to Lemma 2, we present an algorithm computing the $d$ most frequent bigrams, but now with the word-packed search of Lemma 5.

**Lemma 6.** *Given an integer d with $d \geq 1$, we can compute the frequencies of the d most frequent bigrams in a text of length n whose characters are drawn from an alphabet of size $\sigma$ in $\mathcal{O}(n^2 \lg \lg \lg n / \log_\sigma n)$ time using $d \lceil \lg(\sigma^2 n/2) \rceil + \mathcal{O}(\lg n)$ bits.*

**Proof.** We allocate a frequency table $F$ of length $d$. For each text position $i$ with $1 \leq i \leq n - 1$, we compute the frequency of $T[i]T[i+1]$ in $\mathcal{O}(n \lg \lg \lg n / \log_\sigma n)$ time with Lemma 5. After computing a frequency, we insert it into $F$ if it is one of the $d$ most frequent bigrams among the bigrams we have already computed. We can perform the insertion in $\mathcal{O}(\lg d)$ time if we sort the entries of $F$ by their frequencies, obtaining $\mathcal{O}((n \lg \lg \lg n / \log_\sigma n + \lg d)n)$ total time. □

| Operation | Description | Example |
|---|---|---|
| input $S$ | | `bbdbbdcbbbdbb` $= S$ |
| $X \leftarrow \text{find}(\texttt{b}, S)$ | search b in $S$ | `1101100111011` $\rightarrow X$ |
| $X \leftarrow \text{rmPreRun}(X)$ | erase prefix of b's | `0001100111011` $\rightarrow X$ |
| $M \leftarrow \text{rmSufRun}(X)$ | erase suffix of b's | `0001100111000` $\rightarrow M$ |
| $B \leftarrow \text{findBigram}(\texttt{01}, M) \,\&\, M$ | starting of each b run | `0001000100000` $\rightarrow B$ |
| $E \leftarrow \text{findBigram}(\texttt{10}, M) \,\&\, M$ | end of each b run | `0000100001000` $\rightarrow E$ |
| $M \leftarrow M \,\&\, \neg B$ | trim head of runs | `0000100011000` $\rightarrow M$ |
| $X \leftarrow B - (E \,\&\, (\texttt{01})^{q/2})$ | bit mask for all runs ending at even positions | `0001000100000` $= B$ $-$ (`0000100001000`& `0101010101010`) $=$ `0001000011000` $\rightarrow X$ |
| $X \leftarrow M \,\&\, X$ | occurrences of all bs belonging to runs ending at even positions | `0001000011000` $= X$ & `0000100011000` $= M$ $=$ `0000000011000` $\rightarrow X$ |
| $\text{popcount}(X \,\&\, (\texttt{01})^{q/2})$ | frequency of all bbs belonging to runs ending at even positions | `0000000011000` $= X$ & `0101010101010` $=$ `0000000001000` |
| $X \leftarrow B - (E \,\&\, (\texttt{10})^{q/2})$ | bit mask for all runs ending at odd positions | `0001000100000` $= B$ $-$ (`0000100001000`& `1010101010101`) $=$ `0000100100000` $\rightarrow X$ |
| $X \leftarrow M \,\&\, X$ | occurrences of all bs belonging to runs ending at odd positions | `0000100100000` $= X$ & `0000100011000` $= M$ $=$ `0000100000000` $\rightarrow X$ |
| $\text{popcount}(X \,\&\, (\texttt{10})^{q/2})$ | frequency of all bbs belonging to runs ending at odd positions | `0000100000000` $= X$ & `1010101010101` $=$ `0000100000000` |

**Figure 5.** Finding a bigram bb in a string $S$ of bit length $q$, where $q$ is the largest multiple of $2\lceil \lg \sigma \rceil$ fitting into a computer word, divided by $\lceil \lg \sigma \rceil$. In the example, we represent the strings $M$, $B$, $E$, and $X$ as arrays of integers with bit width $x := \lceil \lg \sigma \rceil$ and write 1 and 0 for $1^x$ and $0^x$, respectively. Let $\text{findBigram}(\texttt{bc}, X) := \text{find}(\texttt{bc}, X) \mid \text{find}(\texttt{bc}, \texttt{d}X)$ for $\texttt{d} \neq \texttt{b}$ be the frequency of a bigram bc with $\texttt{b} \neq \texttt{c}$ as described in Section 3.1, where the function find returns the output described in Figure 4. Each of the popcount queries gives us one occurrence as a result (after dividing the returned number by $\lceil \lg \sigma \rceil$), thus the frequency of bb in $S$, without looking at the borders of $S$, is two. As a side note, modern computer architectures allow us to shrink the $0^x$ or $1^x$ blocks to single bits by instructions like `_pext_u64` taking a single CPU cycle.

Studying the final time bounds of Equation (1) for the sequential algorithm of Section 2, we see that we spend $\mathcal{O}(n^2)$ time in the first turn, but spend less time in later turns. Hence, we want to run the bit-parallel algorithm only in the first few turns until $f_k$ becomes so large that the benefits of running Lemma 2 outweigh the benefits of the bit-parallel approach of Lemma 6. In detail, for the $k$-th round, we set $d := f_k$ and run the algorithm of Lemma 6 on the current text if $d$ is sufficiently small, or otherwise the algorithm of Lemma 2. In total, we obtain:

$$\mathcal{O}\left(\sum_{k=0}^{\mathcal{O}(\lg n)} \min\left(\frac{n-f_k}{f_k}n\lg f_k, \frac{(n-f_k)^2\lg\lg\lg n}{\log_\tau n}\right)\right) = \mathcal{O}\left(n^2\sum_{k=0}^{\lg n}\min\left(\frac{k}{\gamma^k}, \frac{\lg\lg\lg n}{\log_\tau n}\right)\right)$$

$$= \mathcal{O}\left(\frac{n^2\lg\log_\tau n\lg\lg\lg n}{\log_\tau n}\right) \text{ time in total,} \tag{3}$$

where $\tau = \sigma_m$ is the sum of the alphabet size $\sigma$ and the number of non-terminals, and $k/\gamma^k > \lg\lg\lg n/\log_\tau n \Leftrightarrow k = \mathcal{O}(\lg(\lg n/(\lg\tau\lg\lg\lg n)))$.

To obtain the claim of Theorem 2, it is left to show that the $k$-th round with the bit-parallel approach uses $\mathcal{O}(n^2\lg\lg\lg n/\log_\tau n)$ time, as we now want to charge each text position with $\mathcal{O}(n/\log_\tau n)$ time with the same amortized analysis as after Equation (1). We target $\mathcal{O}(n/\log_\tau n)$ time for:

(1)     replacing all occurrences of a bigram,
(2)     shifting freed up text space to the right,
(3)     finding the bigram with the highest or lowest frequency in $F$,
(4)     updating or exchanging an entry in $F$, and
(5)     looking up the frequency of a bigram in $F$.

Let $x := \lceil\lg\sigma_{i+1}\rceil$ and $q$ be the largest multiple of $x$ fitting into a computer word, divided by $x$. For Item (1), we partition $T$ into substrings of length $q$ and apply Item (1) to each such substring $S$. Here, we combine the two bit vectors of Figure 5 used for the two popcount calls by a bitwise OR and call the resulting bit vector $Y$. Interpreting $Y$ as an array of integers of bit width $x$, $Y$ has $q$ entries, and it holds that $Y[i] = 2^x - 1$ if and only if $S[i]$ is the second character of an occurrence of the bigram we want to replace. (Like in Item (1), the case in which the bigram crosses a boundary of the partition of $T$ is handled individually). We can replace this character in all marked positions in $S$ by a non-terminal $X_{i+1}$ using $x$ bits with the instruction $(S\mathbin{\&}\neg Y)\mathbin{|}((Y\mathbin{\&}L^q)\cdot X_{i+1})$, where $L$ with $|L| = x$ is the bit vector having marked only the least significant bit. Subsequently, for Item (2), we erase all characters $S[i]$ with $Y[i+1] = (Y \ll x)[i] = 2^x - 1$ and move them to the right of the bit chunk $S$ sequentially. In the subsequent bit chunks, we can use word-packed shifting. The sequential bit shift costs $\mathcal{O}(|S|) = \mathcal{O}(\log_{\sigma_{i+1}} n)$ time, but on an amortized view, a deletion of a character is done at most once per original text position.

For the remaining points, our trick is to represent $F$ by a minimum and a maximum heap, both realized as array heaps. For the space increase, we have to lower $\alpha_i$ (and $\gamma_i$) adequately. Each element of an array heap stores a frequency and a pointer to a bigram stored in a separate array $B$ storing all bigrams consecutively. A pointer array $P$ stores pointers to the respective frequencies in both heaps for each bigram of $B$. The total data structure can be constructed at the beginning of the $k$-th round in $\mathcal{O}(f_k)$ time and hence does not worsen the time bounds. While $B$ solves Item (5), the two heaps with $P$ solve Items (3) and (4) even in $\mathcal{O}(\lg f_k)$ time.

In the case that we want to store the output in working space, we follow the description of Section 2.4, where we now use word-packing to find the second occurrence of a bigram in $T_i$ in $\mathcal{O}(n/\log_{\sigma_i} n)$ time.

## 4. Computing MR-Re-Pair in Small Space

We can adapt our algorithm to compute the MR-Re-Pair grammar scheme proposed by Furuya et al. [18]. The difference to Re-Pair is that MR-Re-Pair replaces the most frequent maximal repeat instead of the most frequent bigram, where a maximal repeat is a reoccurring substring of the text whose frequency decreases when extending it to the left or to the right. (Here, we naturally extended the definition of *frequency* from bigrams to substrings meaning the number of non-overlapping occurrences.) Our idea is to exploit the fact that a most frequent bigram corresponds to a most frequent maximal repeat ([18], Lemma 2). This means that we can find a most frequent maximal repeat by extending

all occurrences of a most frequent bigram to their left and to their right until all are no longer equal substrings. Although such an extension can be time consuming, this time is amortized by the number of characters that are replaced on creating an MR-Re-Pair rule. Hence, we conclude that we can compute MR-Re-Pair in the same space and time bounds as our algorithms (Theorems 1 and 2) computing the Re-Pair grammar.

## 5. Parallel Algorithm

Suppose that we have $p$ processors on a concurrent read concurrent write (CRCW) machine, supporting in particular parallel insertions of elements and frequency updates in a frequency table. In the parallel setting, we allow us to spend $\mathcal{O}(p \lg n)$ bits of additional working space such that each processor has an extra budget of $\mathcal{O}(\lg n)$ bits. In our computational model, we assume that the text is stored in $p$ parts of equal lengths, which we can achieve by padding up the last part with dummy characters to have $n/p$ characters for each processor, such that we can enlarge a text stored in $n \lg \sigma$ bits to $n(\lg \sigma + 1)$ bits in $\max(1, n/p)$ time without extra memory. For our parallel variant computing Re-Pair, our working horse is a parallel sorting algorithm:

**Lemma 7** ([37])**.** *We can sort an array of length n in $\mathcal{O}(\max(n/p, 1) \lg^2 n)$ parallel time with $\mathcal{O}(p \lg n)$ bits of working space. The work is $\mathcal{O}(n \lg^2 n)$.*

The parallel sorting allows us to state Lemma 2 in the following way:

**Lemma 8.** *Given an integer d with $d \geq 1$, we can compute the frequencies of the d most frequent bigrams in a text of length n whose characters are drawn from an alphabet of size $\sigma$ in $\mathcal{O}(\max(n, d) \max(n/p, 1) \lg^2 d/d)$ time using $2d \lceil \lg(\sigma^2 n/2) \rceil + \mathcal{O}(p \lg n)$ bits. The work is $\mathcal{O}(\max(n, d) n \lg^2 d/d)$.*

**Proof.** We follow the computational steps of Lemma 2, but (a) divide a scan into $p$ parts, (b) conduct a scan in parallel but a binary search sequentially, and (c) use Lemma 7 for the sorting. This gives us the following time bounds for each operation:

| Operation | Lemma 2 | Parallel |
| --- | --- | --- |
| fill $F'$ with bigrams | $\mathcal{O}(d)$ | $\mathcal{O}(\max(d/p, 1))$ |
| sort $F'$ lexicographically | $\mathcal{O}(d \lg d)$ | $\mathcal{O}(\max(d/p, 1) \lg^2 d)$ |
| compute frequencies of $F'$ | $\mathcal{O}(n \lg d)$ | $\mathcal{O}(n/p \lg d)$ |
| merge $F'$ with $F$ | $\mathcal{O}(d \lg d)$ | $\mathcal{O}(\max(d/p, 1) \lg^2 d)$ |

The $\mathcal{O}(n/d)$ merge steps are conducted in the same way, yielding the bounds of this lemma. □

In our sequential model, we produce $T_{i+1}$ by performing a left shift of the gained space after replacing all occurrences of a most frequent bigram with a new non-terminal $X_{i+1}$ such that we accumulate all free space at the end of the text. As described in our computational model, our text is stored as a partition of $p$ substrings, each assigned to one processor. Instead of gathering the entire free space at $T$'s end, we gather free space at the end of each of these substrings. We bookkeep the size and location of each such free space (there are at most $p$ many) such that we can work on the remaining text $T_{i+1}$ like it would be a single continuous array (and not fragmented into $p$ substrings). This shape allows us to perform the left shift in $\mathcal{O}(n/p)$ time, while spending $\mathcal{O}(p \lg n)$ bits of space for maintaining the locations of the free space fragments.

For $p \leq n$, exchanging Lemma 2 with Lemma 8 in Equation (1) gives:

$$\mathcal{O}\left(\sum_{k=0}^{\mathcal{O}(\lg n)} \frac{n - f_k}{f_k} \frac{n}{p} \lg^2 f_k\right) = \mathcal{O}\left(\frac{n^2}{p} \sum_{k}^{\lg n} \frac{k^2}{\gamma^k}\right) = \mathcal{O}\left(\frac{n^2}{p}\right) \text{ time in total.}$$

It is left to provide an amortized analysis for updating the frequencies in $F$ during the $i$-th turn. Here, we can charge each text position with $\mathcal{O}(n/p)$ time, as we have the following time bounds for each operation:

| Operation | Sequential | Parallel |
|---|---|---|
| linearly scan $F$ | $\mathcal{O}(f_k)$ | $\mathcal{O}(f_k/p)$ |
| linearly scan $T_i$ | $\mathcal{O}(n_i)$ | $\mathcal{O}(n_i/p)$ |
| sort $D$ with $h = |D|$ | $\mathcal{O}(h \lg h)$ | $\mathcal{O}(\max(1, h/p) \lg^2 h)$ |

The first operation in the above table is used, among others, for finding the bigram with the lowest or highest frequency in $F$. Computing the lowest or highest frequency in $F$ can be done with a single variable pointing to the currently found entry with the lowest or highest frequency during a parallel scan thanks to the CRCW model. In the concurrent read exclusive write (CREW) model, concurrent writes are not possible. A common strategy lets each processor compute the entry of the lowest or highest frequency within its assigned range in $F$, which is then merged in a tournament tree fashion, causing $\mathcal{O}(\lg p)$ additional time.

**Theorem 3.** *We can compute Re-Pair in $\mathcal{O}(n^2/p)$ time with $p \leq n$ processors on a CRCW machine with $\max((n/c) \lg n, n\lceil \lg \tau \rceil) + \mathcal{O}(p \lg n)$ bits of working space including the text space, where $c \geq 1$ is a fixed constant and $\tau = \sigma_m$ is the sum of the alphabet size $\sigma$ and the number of non-terminal symbols. The work is $\mathcal{O}(n^2)$.*

## 6. Computing Re-Pair in External Memory

This part is devoted to the first external memory (EM) algorithms computing Re-Pair, which is another way to overcome the memory limitation problem. We start with the definition of the EM model, present an approach using a sophisticated heap data structure, and another approach adapting our in-place techniques.

For the following, we use the EM model of Aggarwal and Vitter [38]. It features fast internal memory (IM) holding up to $M$ data words and slow EM of unbounded size. The measure of the performance of an algorithm is the number of input and output operations (I/Os) required, where each I/O transfers a block of $B$ consecutive words between memory levels. Reading or writing $n$ contiguous words from or to disk requires $\text{scan}(n) = \Theta(n/B)$ I/Os. Sorting $n$ contiguous words requires $\text{sort}(n) = \mathcal{O}((n/B) \cdot \log_{M/B}(n/B))$ I/Os. For realistic values of $n$, $B$, and $M$, we stipulate that $\text{scan}(n) < \text{sort}(n) \ll n$.

A simple approach is based on an EM heap maintaining the frequencies of all bigrams in the text. A state-of-the-art heap is due to Jiang and Larsen [39] providing insertion, deletion, and the retrieval of the maximum element in $\mathcal{O}(B^{-1} \log_{M/B}(N/B))$ I/Os, where $N$ is the size of the heap. Since $N \leq n$, inserting all bigrams takes at most $\text{sort}(n)$ I/Os. As there are at most $n$ additional insertions, deletions, and maximum element retrievals, this sums to at most $4 \text{sort}(n)$ I/Os. Given Re-Pair has $m$ turns, we need to scan the text $m$ times to replace the occurrences of all $m$ retrieved bigrams, triggering $m \sum_{i=1}^{m} \text{scan}(|T_i|) \leq m \text{scan}(n)$ I/Os.

In the following, we show an EM Re-Pair algorithm that evades the use of complicated data structures and prioritizes scans over sorting. This algorithm is based on our Re-Pair algorithm. It uses Lemma 2 with $d := \Theta(M)$ such that $F$ and $F'$ can be kept in RAM.

This allows us to perform all sorting steps and binary searches in IM without additional I/O. We only trigger I/O operations for scanning the text, which is done $\lceil n/d \rceil$ times, since we partition $T$ into $d$ substrings. In total, we spend at most $mn/M$ scans for the algorithm of Lemma 2. For the actual algorithm having $m$ turns, we update $F$ $m$ times, during which we replace all occurrences of a chosen bigram in the text. This gives us $m$ scans in total. Finally, we need to reason about $D$, which we create $m$ times. However, $D$ may be larger than $M$, such that we may need to store it in EM. Given that $D_i$ is $D$ in the $i$-th turn, we sort $D$ in EM, triggering $\mathrm{sort}(D_i)$ I/Os. With the converse of Jensen's inequality ([40], Theorem B) (set there $f(x) := n \lg n$), we obtain $\sum_{i=1}^{m} \mathrm{sort}(|D_i|) \le \mathrm{sort}(n) + \mathcal{O}(n \log_{M/B} 2)$ total I/Os for all instances of $D$. We finally obtain:

**Theorem 4.** *We can compute Re-Pair with* $\min(4 \, \mathrm{sort}(n), (mn/M) \, \mathrm{scan}(n) + \mathrm{sort}(n) + \mathcal{O}(n \log_{M/B} 2)) + m \, \mathrm{scan}(n)$ *I/Os in external memory.*

The latter approach can be practically favorable to the heap based approach if $m = o(\lg n)$ and $mn/M = o(\lg n)$, or if the EM space is also of major concern.

## 7. Heuristics for Practicality

The achieved quadratic or near-quadratic time bounds (Theorems 1 and 2) seem to convey the impression that this work is only of purely theoretic interest. However, we provide here some heuristics, which can help us to overcome the practical bottleneck at the beginning of the execution, where only $\mathcal{O}(\lg n)$ of bits of working space are available. In other words, we want to study several heuristics to circumvent the need to call Lemma 2 with a small parameter $d$, as such a case means a considerable time loss. Even a single call of Lemma 2 with a small $d$ prevents the computation of Re-Pair of data sets larger than 1 MiB within a reasonable time frame (cf. Section 2.6). We present three heuristics depending on whether our space budget on top of the text space is within:

1.  $\sigma_i^2 \lg n_i$ bits,
2.  $n_i \lg(\sigma_{i+1} + n_i)$ bits, or
3.  $\mathcal{O}(\lg n)$ bits.

**Heuristic 1.** If $\sigma_i$ is small enough such that we can spend $\sigma_i^2 \lg n_i$ bits, then we can count the frequencies of all bigrams in a table of $\sigma_i^2 \lg n_i$ bits in $\mathcal{O}(n)$ time. Whenever we reach a $\sigma_j$ that lets $\sigma_j \lg n_j$ grow outside of our budget, we have spent $\mathcal{O}(n)$ time in total for reaching $T_j$ from $T_i$ as the costs for replacements can be amortized by twice of the text length.

**Heuristic 2.** Suppose that we are allowed to use $(n_i - 1) \lg(n_i/2) = (n_i - 1) \lg n_i - n_i + \mathcal{O}(\lg n_i)$ bits in addition to the $n_i \lg \sigma_i$ bits of the text $T_i$. We create an extra array $F$ of length $n_i - 1$ with the aim that $F[j]$ stores the frequency of $T[j]T[j+1]$ in $T[1..j]$. We can fill the array in $\sigma_i$ scans over $T_i$, costing us $\mathcal{O}(n_i \sigma_i)$ time. The largest number stored in $F$ is the most frequent bigram in $T$.

**Heuristic 3.** Finally, if the distribution of bigrams is skewed, chances are that one bigram outnumbers all others. In such a case, we can use the following algorithm to find this bigram:

**Lemma 9.** *Given there is a bigram in $T_i$ ($0 \le i \le n$) whose frequency is higher than the sum of frequencies of all other bigrams, we can compute $T_{i+1}$ in $\mathcal{O}(n)$ time using $\mathcal{O}(\lg n)$ bits.*

**Proof.** We use the Boyer–Moore majority vote algorithm [41] for finding the most frequent bigram in $\mathcal{O}(n)$ time with $\mathcal{O}(\lg n)$ bits of working space.  $\square$

A practical optimization of updating $F$ as described in Section 2.3 could be to enlarge $F$ beyond $f_k$ instead of keeping its size. There, after a replacement of a bigram with a non-terminal $X_{i+1}$, we insert those bigrams containing $X_{i+1}$ into $F$ whose frequencies are above $t_k$ while discarding bigrams of the lowest frequency stored in $F$ to keep the size of $F$

at $f_k$. Instead of discarding these bigrams, we could just let $F$ grow. We can let $F$ grow by using the space reserved for the frequency table $F'$ computed in Lemma 2 (remember the definition of the constant $\alpha_i$). By doing so, we might extend the lifespan of a round.

## 8. Conclusions

In this article, we propose an algorithm computing Re-Pair in-place in sub-quadratic time for small alphabet sizes. Our major tools are simple, which allows us to parallelize our algorithm or adapt it in the external memory model.

This paper is an extended version of our paper published in The Prague Stringology Conference 2020 [42] and our poster at the Data Compression Conference 2020 [43].

**Author Contributions:** Conceptualization: K.G.; formal analysis: T.I., I.F., Y.T., and D.K.; visualization: K.S.; writing: T.I. and D.K. All authors read and agreed to the published version of the manuscript.

**Funding:** This work is funded by the JSPS KAKENHI Grant Numbers JP18F18120 (Dominik Köppl), 19K20213 (Tomohiro I), and 18K18111 (Yoshimasa Takabatake) and the JST CREST Grant Number JPMJCR1402 including the AIP challenge program (Keisuke Goto).

**Institutional Review Board Statement:** Not applicable.

**Informed Consent Statement:** Not applicable.

**Data Availability Statement:** Data sharing not applicable.

**Conflicts of Interest:** The authors declare no conflict of interest.

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
