# Peer review of "Re-Pair in Small Space†"

_algorithms, doi:10.3390/a14010005_

Round 1

Reviewer 1 Report

The paper is well structured and presented. Nevertheless, giving a short explanation for acronyms (quite well known to experienced users, but less known to the accidental reader) could increase the interested audience (e.g. line 54, "LCP array").

The algorithm computes Re-Pair in-place in sub-quadratic time for small alphabet sizes. In that case, other compression schemes could be more convenient. A comparison with those ones could explain in more detail your solution compromise.

Furthermore, the proposed algorithm is very slow, taking already one hour for computing Re-Pair on 1 MiB text files. The algorithm described in the paper takes O(n^2) time, at it looks likely that this time bound is tight. These notes should appear in the abstract to provide potential reader with more information on practical implementation of your work.

Reviewer 2 Report

This article proposes, prove correctness and time complexity analysis of an in-place subquadratic time algorithm for Re-pairs on texts (strings) on small alphabets. The article provides empiric and comparative analysis with well-known methods for Re-pairing.

The article is well-written, the mathematical proofs and definitions are sound to me. The Heuristics analysis at the end of the paper is only listed with a lemma (9) to support their adequateness. However, any motivation for the range of application of the heuristics is provided. The article points out to implementation and the experiments conduct with it. This is very good, it is in git-hub. We can compare its performance with well-known implementations to broader minimal grammar generation approaches as IRR-ML and LFS2 , that use suffix-tree. The text would be improved if it could explicitly compare to methods like the above mentioned and the some of their extensions IRCOO, etc. Of course, some of them are not only Re-pair algorithms,  they can deal with more general ways to provide the smallest grammar. How do they compare in the particular case a Re-pair is optimal for generating grammars. Well, at least some mention to that would be very useful.

In page 3, the first line of the second paragraph should be \Omega(n)^2,

Reviewer 3 Report

Re-Pair in Small Space.

The paper presents a new algorithm for computing Re-Pair compression. The main result is an algorithm that uses quadratic time and sublinear space -- essentially n/c words + lower order terms, where c is a constant. The algorithm assumes that the text is a rewriteable part of the working space. Several extension are also studied, including a bit-parallel improvement for special cases, a parallel version, and an external memory version. The algorithm is also implemented and results are presented. The main technical contribution is a simple algorithm for space-efficient computation of frequencies.

The results provide new insights into the problem and I therefore suggest to accept the paper. I'm not convinced that the trade-off is practically relevant due to the quadratic time complexity, but from a theoretical perspective it is interesting that this can be achieved.

Some misc. comments:
- abstract "fix user-defined" -> "fixed user-defined".
- the \cup symbol uses for O-notation should be changed to a min. or you should change "=" to \in or \subseteq in O-notation expressions for consistency.

- l. 87 n -> n^2

- l 115. Avoid overly emotional english such as "Embracing". E.g. just use "Using the frequency tables ..." Check rest of paper for such bugs.

- before eq l. 249. "yield" -> "get". Also replace similar uses of "yield" in the rest of the paper.

- l. 307. The implementation section does not compare with other algorithms. Doing so would be much more interesting.
